# Addition of digital VIA/VILI to conventional naked-eye examination for triage of HPV-positive women: A study conducted in a low-resource setting

Eva Dufeil[1]*, Bruno Kenfack[2,3], Evelyn Tincho[2,4], Jovanny Fouogue[5], Ania Wisniak[6,7], Jessica Sormani[6,8], Pierre Vassilakos[6,9], Patrick Petignat[1,6]

1 Faculty of Medicine, University of Geneva, Geneva, Switzerland, 2 Faculty of Medicine and Pharmaceutical Science, University of Dschang, Dschang, Cameroon, 3 Department of Gynecology and Obstetrics, District Hospital of Dschang, Dschang, Cameroon, 4 Department of Gynecology and Obstetrics, Faculty of Medicine and Biomedical Sciences, Centre Hospitalier Universitaire (CHUY), Yaoundé, Cameroon, 5 Department of Obstetrics and Gynecology, Bafoussam Regional hospital, Bafoussam, Cameroon, 6 Gynecology Division, Department of Gynecology and Obstetrics, University Hospitals of Geneva, Geneva, Switzerland, 7 Population Epidemiology Unit, Department of Primary Care, Geneva University Hospitals, Geneva, Switzerland, 8 School of Health Sciences, HES-SO University of Applied Sciences and Arts Western Switzerland, Geneva, Switzerland, 9 Geneva Foundation for Medical Education and Research, Geneva, Switzerland

* eva-camille.dufeil@etu.unige.ch

**Data Availability Statement:** The data used in this study are available in the Yareta database at DOI: 10.26037/yareta:ysefjw2ofre3fnui2buscyixtu or

## Abstract

### Background

World Health Organization guidelines for cervical cancer screening recommend HPV testing followed by visual inspection with acetic acid (VIA) for triage if HPV positive. In order to improve visual assessment and identification of cervical intraepithelial neoplasia grade 2 and worse (CIN2+), providers may use visual aids such as digital cameras.

### Objectives

To determine whether combined examination by naked-eye and digital VIA (D-VIA) and VILI (D-VILI) improves detection of CIN2+ as compared to the conventional evaluation.

### Materials and methods

Women (30–49 years) living in Dschang (West Cameroon) were prospectively invited to a cervical cancer screening campaign. Primary HPV-based screening was followed by VIA/VILI and D-VIA/VILI if HPV-positive. Health care providers independently defined diagnosis (pathological or non-pathological) based on naked-eye VIA/VILI and D-VIA/VILI. Decision to treat was based on combined examination (VIA/VILI and D-VIA/VILI). Cervical biopsy and endocervical curettage were performed in all HPV-positive participants and considered as reference standard. Diagnostic performance of individual and combined naked-eye VIA/VILI and D-VIA/VILI was evaluated. A sample size of 1,500 women was calculated assuming a prevalence of 20% HPV positivity and 10% CIN2+ in HPV-positive women.

with this link: https://yareta.unige.ch/#/home/detail/ffbeb6d7-b390-4755-987e-8faf85f97c67.

**Funding:** This study was funded by ESTHER Switzerland 17G1, Service de Solidarité Internationale (Canton of Geneva), Hôpitaux Universitaires de Genève (HUG) (Switzerland), the Fondation Privée of HUG, the Commission des Affaires Humanitaires (CAH) of HUG, the Groupement Romand de la Société Suisse de Gynécologie et Obstétrique (GRSSGO), and the University of Dschang (Cameroon). The funders had no role in study design, data collection and analysis, decision to publish, or preparation of the manuscript.

**Competing interests:** The authors have declared that no competing interests exist.

## Results

Due to the COVID-19 pandemic, the study had to terminate prematurely. A total of 1,081 women with a median age of 40 (IQR 35.5–45) were recruited. HPV positivity was 17.4% (n = 188) and 26 (14.4%) had CIN2+. Naked-eye VIA and D-VIA sensitivities were 80.8% (95% CI 60.6–93.4) and 92.0% (95% CI 74.0–99.0), and specificities were 31.2% (95% CI 24–39.1) and 31.6% (95% CI 24.4–39.6), respectively. The combination of both methods yielded a sensitivity of 92.3% (95% CI 74.9–99.1) and specificity of 23.2% (95% CI 16.8–30.7). A trend towards improved sensitivity was observed, but did not reach statistical significance.

## Conclusion

Addition of D-VIA/VILI to conventional naked-eye examination may be associated with improved CIN2+ identification. Further studies including a larger sample size are needed to confirm these results.

## Introduction

Low-income countries like Cameroon have a high prevalence rate of cervical cancer, representing the second leading cause of death by cancer in women [1–3]. In Cameroon, 33.7 per 100'000 women are affected annually, despite cervical cancer being a highly preventable disease [4].

Cervical cancer screening allows to prevent and detect the disease at a precancerous stage and treat it before it progresses to cancer. However, in low-resource settings, the lack of organized screening due, in part, to the absence of dedicated medical structures and trained staff contributes to the problem. Advances in technology for cervical detection in low-resource settings using human papillomavirus (HPV) primary testing followed by immediate treatment with cryotherapy or thermal ablation if needed are adapted to low-resource contexts [5–7] as a 'screen-and-treat' approach in a single visit requires fewer resources and reduces risk of loss to follow-up.

Recently, the World Health Organization (WHO) has led several initiatives towards the elimination of cervical cancer as a public health concern [8]. These initiatives have three main targets in order to eliminate cervical cancer by 2030: (i) to vaccinate 90% of young girls against HPV, (ii) to screen 70% of the target population with a highly sensitive test by age 35 and again by age 45, and (iii) to provide appropriate management to 90% of women identified with cervical disease [9].

One option recommended by the WHO's to prevent cervical cancer in low-income countries is HPV primary screening followed by triage of HPV-positive women using visual assessment [8]. However, interpreting visual inspection with acetic acid and lugol's iodine (VIA/VILI) with the naked eye alone is subjective and can be highly variable between health care providers, an issue that a supportive approach with the use of digital image capture can help to overcome [10–13]. To mitigate some of the weaknesses in successful implementation of VIA, investigators have developed enhanced digital imaging of the cervix (termed cervicography) with a camera in Zambia [14–16]. The method requiring a camera connected with a television screen allowed an improved assessment of VIA [17, 18].

The advent of smartphones with high-performance cameras represents an alternative to colposcopy and an opportunity for innovative clinical methods which can be

implemented with rather inexpensive tools. Smartphones allow digital image acquisition and inspection with acetic acid and lugol's iodine (hereafter referred as D-VIA/VILI) allowing to reveal cervical features which may not be visible with the unaided eye (Fig 1). Smartphone cameras provide an alternative to colposcopy and a reliable option for integrating telemedicine but, to date, it is still undetermined whether a combined approach of naked-eye and digital VIA/VILI performs better than naked-eye examination for the diagnosis of cervical intraepithelial neoplasia (CIN) grade 2 or worse (CIN2+). Sensitivity of D-VIA/VILI varies between studies, partly due to the diagnostic challenge of CIN2 + lesions and the absence of a valid reference standard. Our hypothesis was that the combined approach (naked-eye VIA/VILI and D-VIA/VILI) compared to conventional naked-eye VIA assessment would lead to improvement of CIN2+ detection. The objective of this study was to evaluate the diagnostic accuracy of CIN2+ detection using a combined approach (naked-eye and digital VIA) as compared to a traditional naked-eye approach.

## Material and methods

### Setting and study population

This study was nested in a larger cervical cancer prevention project established 20 years ago in Cameroon with the aim to improve cervical cancer prevention [19]. The study protocol has already been published [20, 21]. The study was conducted in Dschang (West Cameroon) between February 2019 and March 2020. Women aged 30 to 49 years old living in the health district area were invited to participate in a free cervical cancer screening campaign. Exclusion criteria were a positive history of cervical cancer, cervical cancer treatment or hysterectomy, and pregnancy at the time of enrolment.

### Procedure

The screening and (if needed) treatment process took place over one day, as recommended by the WHO (Fig 2) [8]. After providing signed informed consent, participants were administered questionnaires by a midwife on their socio-demographic characteristics, gynecological and obstetrical history, smoking status and HIV status. Participants were then invited to perform HPV self-sampling (Self-HPV) as primary screening followed by on-site analysis using the Xpert HPV assay (GeneXpert; Cepheid, Sunnyvale, CA, USA). In case of a negative HPV test, women were advised to repeat screening in 5 years. If the HPV test was positive, a pelvic exam was performed with a PAP-test, followed by triage by VIA/VILI, and finally sampling by endocervical brushing and a biopsy at the site of pathological VIA/VILI or at 6 o'clock when VIA/VILI was negative. The cervical exam was performed by certified midwives. Midwives independently performed a naked-eye cervical evaluation: native (i.e., product-free visualization of the cervix), after application of acetic acid and Lugol's iodine. A second midwife independently captured images during the pelvic exam (one native, one after application of acetic acid and one after application of Lugol's iodine) using a smartphone (Samsung Galaxy J5, Seoul, Korea). The smartphone allowed more time for a detailed inspection of the photo, to magnify the lesion by zooming in, and to compare native, acetic acid and iodine colorations by sliding through pictures, thus potentially improving detection of abnormalities.

VIA/VILI and D-VIA/VILI were independently interpreted as non-pathological (no precancerous lesion) or pathological (suspicious of a precancerous or cancerous lesion). Decision to treat was based on the combined results (VIA/VILI and D-VIA/VILI); if one of the two tests (VIA/VILI or D-VIA/VILI) was positive, the decision was to treat. If the final decision was to

(1) Cervix without precancerous lesion. Left. Native Cervix. In the center. Cervix after acetic acid application (VIA). Right. Cervix after Lugol's iodine application (VIA).
(2) Cervix with precancerous lesion. Left. Native Cervix. In the center. Cervix after acetic acid application (VIA). Right. Cervix after Lugol's iodine application (VIA).

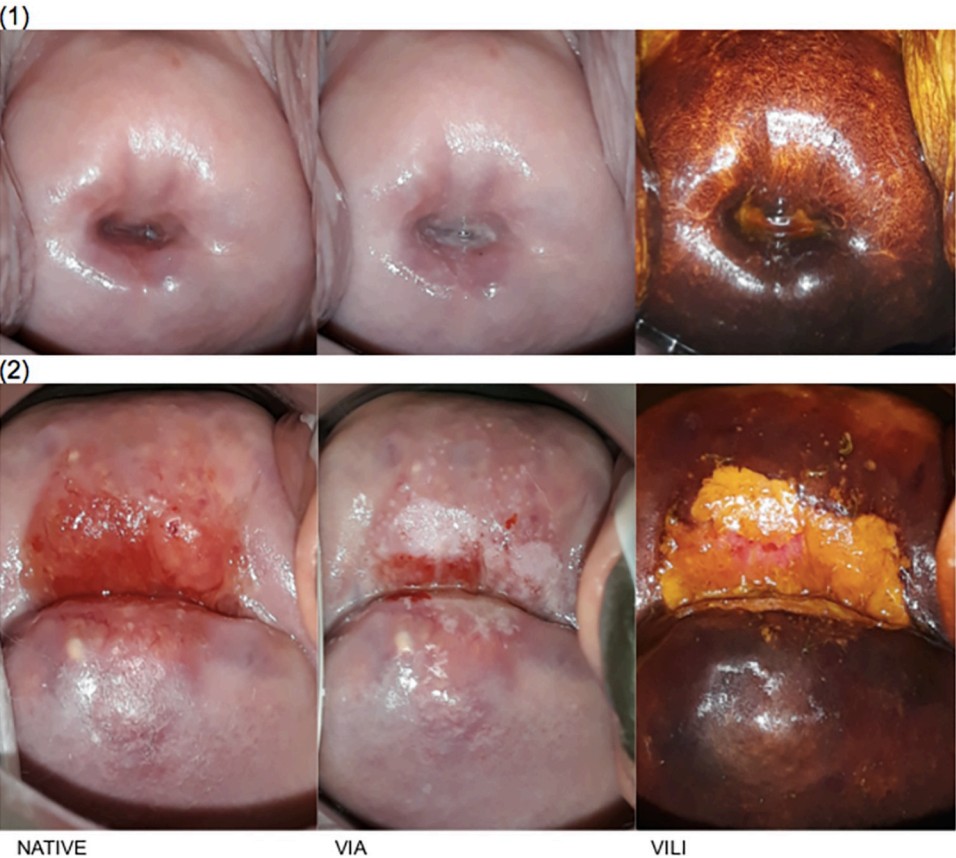

NATIVE        VIA        VILI

**Fig 1. Digital visualization of a healthy (1) and pathological (2) cervix using acetic acid (VIA) and Lugol's iodine (VILI).**

treat, women were immediately treated by thermal ablation if eligible, or referred for further evaluation if non-eligible for thermal ablation.

Smartphone photography allowed easy storage and retrieval of VIA images for quality control and educational purposes. Supervision of the screening activity was performed twice a month by a gynecologist specialized in colposcopy who reviewed all photos with the frontline health care providers. Patients who were undertreated according to the specialist's diagnosis, as well as those with CIN2+ on histological analysis which were not identified through VIA/VILI or D-VIA/VILI, were called back for treatment. If the final decision was not to treat, women were invited for a follow-up in 1 year. The study data collection was conducted by midwives under the supervision of an on-site research assistant through standardized case report forms, and later transcribed to a web-based electronic database (secuTRial®). Smartphone images were anonymously stored on the server of the Geneva University Hospitals in a password-protected computer.

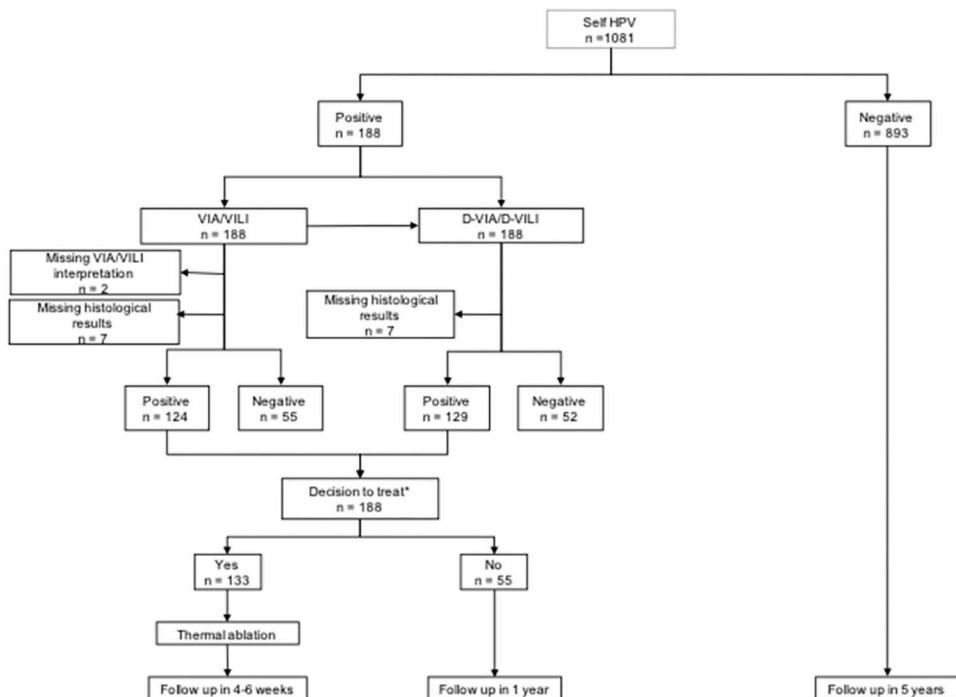

**Fig 2. Flowchart of the study.** * if one of the two tests (VIA or D-VIA) was positive, the decision was to treat.
**Abbreviations:** Self HPV = HPV self-sampling, NA = not analysable, VIA = visual inspection of the cervix with acetic acid; VILI = visual inspection of the cervix with Lugol's iodine.

## VIA/VILI interpretation

A lower threshold for VIA positivity than is usually recommended by the International Agency for Research on Cancer criteria was applied [22]. ABCD criteria as previously reported were used for diagnosis [23]. Briefly, we considered any acetowhite lesion (faint, translucent, or dense) larger than 5 mm touching the cervical transformation zone, including those that were indeterminate or uncertain, to be positive (or pathological). We considered any lesion revealed by iodine that "matched" an acetowhite lesion to be a positive VILI.

## Reference standard

A cervical biopsy and endocervical curettage (ECC) were performed on all participants having a positive HPV test. This material was analyzed at the University Hospitals of Geneva pathology laboratory (the pathology laboratory at Hospital of Gynecology, Obstetrics and Pediatrics of Yaounde was temporary closed due to renovations). Histological results, used as the reference standard, were classified by cervical intraepithelial neoplasm grade. A negative result was considered a cervix without CIN or a CIN1. Positive results included CIN2 or worse, including CIN2, CIN3, adenocarcinoma in-situ (AIS) and cancer.

## Ethical considerations

The protocol has obtained approval from the Cantonal Ethics Board of Geneva, Switzerland (Commission cantonale d'éthique de la recherche, N˚2017–01110 and N˚2020–00868) and the

**Table 1. Baseline sociodemographic characteristics and reproductive health indicators.**

| Variable | Total |
|---|---|
| Participants recruited, n | 1081 |
| HPV status, n (%) | |
| Positive | 188 (17.4) |
| Negative | 893 (82.6) |
| Age (years), median (IQR) | 40 (35–45) |
| Marital status, n (%) | |
| Single | 87 (8.1) |
| With partner | 449 (41.6) |
| Married | 458 (42.5) |
| Divorced/widowed | 85 (7.9) |
| Education, n (%) | |
| Unschooled | 7 (0.7) |
| Primary education | 320 (29.7) |
| Secondary education | 580 (53.8) |
| Tertiary education | 171 (15.9) |
| Other | 1 (0.1) |
| Employment status, n (%) | |
| Employed without responsibility | 239 (22.2) |
| Employed with responsibility | 34 (3.2) |
| Self-employed | 296 (27.4) |
| Housewife | 244 (22.6) |
| Farmer | 226 (21.0) |
| Other | 40 (3.7) |
| Age at menarche (years), mean ± SD | 14.7 (±1.8) |
| Age at first intercourse, median (IQR) | 18 (16–19) |
| Number of sexual partners, median (IQR) | 3 (2–5) |
| Contraception, n (%) | |
| None | 786 (72.7) |
| Condom | 86 (8.0) |
| Hormonal contraception | 184 (17.0) |
| Other | 8 (0.7) |
| Unknown | 4 (0.4) |
| HIV status, n (%) | |
| Negative | 1019 (94.3) |
| Positive | 36 (3.3) |
| Unknown | 26 (2.4) |
| Previous HPV infection, n (%) | |
| Negative | 901 (83.4) |
| Positive | 178 (16.5) |
| Treated | 23 (2.1) |
| Unknown | 2 (0.2) |
| Age at first delivery (years), mean ± SD | 21.0 (±5.2) |
| Parity, n (%) | |
| Nulliparous | 44 (4.1) |
| 1–4 | 467 (43.3) |
| >4 | 568 (52.6) |
| Gravidity, n (%) | |

(*Continued*)

**Table 1.** (Continued)

| Variable | Total |
|---|---|
| Nulligravida | 25 (2.3) |
| 1–4 | 315 (29.2) |
| >4 | 739 (68.5) |
| Tobacco | |
| Non-smoking | 1057 (98.1) |
| Smoking | 21 (1.9) |

**Abbreviations:** IQR = interquartile range; SD = standard deviation; HIV = human immunodeficiency virus; HPV = human papillomavirus.

National Ethics Committee for Research on Human Health, Cameroun (Comité national d'éthique de la recherche pour la santé humaine, CNERSH, N˚2018/07/1083/CE/CNERSH/ SP). The study protocol was registered under ClinicalTrials.gov (number NCT03757299). All participants provided informed written consent before enrolment in the study.

## Statistical analysis

We estimated the minimum sample size required based on an expected difference of sensitivity of 20% between the two methods, with a power of 80% to detect it and a two-sided level of significance of 0.05. Considering an estimated 20% of positive HPV tests and 10% of CIN2+ lesions among HPV-positive women (estimated from our previous experiences in Cameroon), we planned to recruit 1,500 participant, using the formula $n = (Z\alpha/2+Z\beta)2 * (p1(1-p1)+p2(1-p2)) / (p1-p2)2$, where n is the sample size, $Z\alpha/2$ is the critical value of the normal distribution at $\alpha/2$, $Z\beta$ is the critical value of the normal distribution at $\beta$ and p1 and p2 are the expected sample proportions of the two groups. However, due the COVID-19 pandemic, screening activities at the clinical site were suspended in March 2020. Therefore, we had to consider this analysis with a smaller than expected sample size. The sensitivity and specificity of each method (naked-eye VIA/VILI vs combined naked-eye and D-VIA/VILI) were calculated based on intention to treat, using the histological results as the reference standard. The positivity of each method was also determined as well as positive and negative predictive values (PPV and NPV). P-values comparing sensitivities and specificities between methods were estimated using McNemar's Chi$^2$ test, and p-values for NPV and PPV using a generalized score statistic for paired designs [24].

## Results

### Participants' characteristics

During the period under study, 1081 women (median age 40, IQR 35–45) were recruited for HPV-sampling, with an HPV positivity rate of 17.4%. One hundred and eighty-eight HPV-positive women underwent a pelvic exam for visual assessment. The HPV genotypes identified were HPV-16 (4.3%), HPV-18/45 (8.0%), and other high-risk HPV types pooled together (77.1%). Eighteen women were infected with more than one HPV, including HPV-16 and 18 (1.6%), 16 and other (4.3%), or 18 and other (3.7%). No woman had 3 different types of HPV. Table 1 reports the participants' sociodemographic and clinical data.

### Histological results

Among 188 biopsies, 26 (13.8%) were pathological (CIN2+), 155 non-pathological (82.4%), which were either negative or CIN1, and 7 (3.7%) could not be analyzed because of insufficient

**Table 2. VIA/VILI, D-VIA/VILI and combined assessment of histopathologic results, categorized by CIN grade.**

| CIN grade n (%) | VIA/VILI | | D-VIA/VILI | | COMBINED VIA/VILI & D-VIA/VILI | |
|---|---|---|---|---|---|---|
| | Positive | Negative | Positive | Negative | Positive | Negative |
| **Negative\*** | 73 (64.6) | 40 (35.4) | 72 (63.2) | 42 (36.8) | 82 (71.9) | 32 (28.1) |
| **CIN1** | 33 (80.5) | 8 (19.5) | 34 (82.9) | 7 (17.1) | 37 (90.2) | 4 (9.8) |
| **CIN2\*\*** | 9 (81.8) | 2 (18.2) | 9 (90.0) | 1 (10.0) | 10 (90.9) | 1 (9.1) |
| **CIN3** | 11 (78.6) | 3 (21.4) | 13 (92.6) | 1 (7.1) | 13 (92.6) | 1 (7.1) |
| **Cancer** | 1 (100) | 0 | 1 (100) | 0 | 1 (100) | 0 |
| **Total CIN2+** | 21 (80.8) | 5 (19.2) | 23 (92.0) | 2 (8.0) | 24 (92.3) | 2 (7.7) |

Abbreviations: CIN = cervical intraepithelial neoplasm; VIA = visual inspection of the cervix with acetic acid; VILI = visual inspection of the cervix with Lugol's iodine;
DVIA = digital visual inspection of the cervix with acetic acid; DVILI = digital visual inspection of the cervix with Lugol's iodine.

\*1 missing data for VIA/VILI.

\*\*1 non-interpretable D-VIA/VILI.

material. In these 26 pathological lesions, there were 11 CIN2, 14 CIN3 and one cancerous lesion. Table 2 shows the results of visual assessment according to histology.

### VIA/VILI and D-VIA/VILI diagnostic accuracy

If either VIA/VILI or D-VIA/VILI was positive the combined visual assessment was considered positive. Note that patient 11 in the Table is reported as positive after VIA/VILI assessment but was not treated as the D-VIA/VILI was of insufficient quality for assessment, due to the presence of blood. D-VIA/VILI identified three patients with precancerous lesions that VIA/VILI had missed. Both patients with CIN2+ identified by histological analysis who were missed by VIA and D-VIA were called back and subsequently treated. Out of 188 participants, there were 187 results of VIA/VILI and 187 of D-VIA/VILI. VIA's positivity was 69.4% and that of D-VIA/VILI 72.4%. VIA/VILI's sensitivity was 80.8% (95% CI 60.6–93.4), specificity was 31.2% (95% CI 24–39.1), PPV was 16.5% (95% CI 10.5–24.2) and NPV was 90.6% (95% CI 79.3–96.9). For D-VIA/VILI, sensitivity was 92.0% (95% CI 74.0–99.0, p = 0.833 with VIA/VILI as reference group), specificity was 31.6% (95% CI 24.4–39.6, p = 0.8415), PPV was 17.8% (95% CI 11.7–25.5, p = 0.3268) and NPV was 96.1% (95% CI 86.5–99.5, p = 0.0771). As for the combination of VIA/VILI and D-VIA, sensitivity was 92.3% (95% CI 74.9–99.1, p = 0.0833 with VIA/VILI as reference group), specificity 23.2% (95% CI 16.8–30.7, p = 0.0005), PPV 16.8% (95% CI 11.1–23.9, p = 0.7500) and NPV 94.7% (95% CI 82.3–99.4, p = 0.1963). Table 3 summarizes these results. Considering our final sample of 26 CIN2+ participants, the current study was powered at 43.6% for the detection of a 20% difference between the sensitivities of VIA/VILI and the combined VIA/VILI and D-VIA/VILI method.

### Discussion

Triaging of HPV-positive women with cytology or biomarkers is not readily implementable in low-resource settings. Therefore, alternative methods like VIA have been considered. Some reports found that VIA used as a triage method is associated with an important loss of sensitivity and may be inappropriate for use in a triaging strategy [25, 26]. The main reason is that interpreting VIA by naked eye alone is a highly subjective method and widely depends on the experience of the provider [27–29]. In this project, we evaluated if acquisition of digital cervical images (native, after VIA and VILI) with a smartphone has the potential to improve traditional unaided cervical visual assessment to detect precancerous lesions. Camera or smartphone-enhanced VIA/VILI represent an innovative method and a pragmatic alternative to

**Table 3. Sensitivity, specificity, PPV and NPV of each diagnostic method.**

| Diagnostic Method | Sensitivity % (95% CI) | Sensitivity's p-value (vs VIA/VILI) | Specificity % (95% CI) | Specificity's p-value (vs VIA/VILI) | PPV % (95% CI) | PPV's p-value (vs VIA/VILI) | NPV % (95% CI) | NPV's p-value (vs VIA/VILI) |
|---|---|---|---|---|---|---|---|---|
| VIA/VILI | 80.8 (60.6–93.4) | | 31.2 (24–39.1) | | 16.5 (10.5–24.2) | | 90.6 (79.3–96.9) | |
| D-VIA/D-VILI | 92.0 (74.0–99.0) | 0.0833 | 31.6 (24.4–39.6) | 0.8415 | 17.8 (11.7–25.5) | 0.3268 | 96.1 (86.5–99.5) | 0.0771 |
| VIA/VILI and D-VIA/D-VILI combined | 92.3 (74.9–99.1) | 0.0833 | 23.2 (16.8–30.7) | 0.0005 | 16.8 (11.1–23.9) | 0.7500 | 94.7 (82.3–99.4) | 0.1963 |

**Abbreviations:** VIA = visual inspection of the cervix with acetic acid; VILI = visual inspection of the cervix with Lugol's iodine; PPV = positive predictive value; NPV = negative predictive value; D-VIA = digital visual inspection of the cervix with acetic acid; D-VILI = digital visual inspection of the cervix with Lugol's iodine.

colposcopy adapted to low-resource contexts [18]. However, the major question is to determine if D-VIA/VILI has an impact on the clinical management of HPV-positive women as compared to naked-eye evaluation. A critical issue for the patient and the provider is, first of all, to reduce the risk of missing a CIN2+ lesion, rather than improving diagnostic accuracy.

To the best of our knowledge, this is the first prospective trial which aimed to investigate the impact of the adjunct of D-VIA/VILI to conventional naked-eye examination. Previous studies on the possible role of camera-enhanced diagnosis of cervical precancerous lesions and monitoring with digital equipment have been reported, with sensitivity and specificity of D-VIA ranging between 66.7% and 94.1%, and 50.4% and 85.4% respectively [17, 30], which is consistent with our results. However, little is known about the addition of digital images to the final decision regarding precancerous lesion management in low-resource settings [10, 17].

Our study supports that the addition of smartphone-enhanced VIA/VILI may lead to improvement in CIN2+ detection, although only borderline statistical significance was reached in regard to sensitivity. Considering D-VIA/VILI individually, we also observed a trend in sensitivity improvement. Its positive and negative predictive values were also slightly higher, however without reaching statistical significance. On the other hand, digital assessment reduced specificity compared to the naked eye alone. A loss in specificity may however be considered acceptable in low-resource contexts where loss to follow-up is a common issue, and considering the availability of safe and well-tolerated therapeutic options such as thermal ablation. The combined use of VIA/VILI and D-VIA/VILI further allowed to detect two patients with a CIN3 lesion, and one patient with a CIN2 lesion.

The inclusion of a digital approach to cervical screening strategies could be a good adjunct to VIA/VILI. The combination of a conventional naked-eye approach with a digital approach has already been performed in dermatology for the detection of melanomas and showed an improvement in performance [31]. Although not a scientific validation of the technique, in daily practice, we observed that front-line providers having experience with D-VIA/VILI acquisition unanimously prefer the use of digital images, simply because it offers the possibility to be manipulated to zoom in on suspicious regions or transformation zones, as well as the possibility to simultaneously compare native, VIA and VILI images and ask colleagues for a second opinion in real time. In cases of doubt, D-VIA/VILI could be verified remotely by a more experienced person [32].

The main limitation of the study is the sample size which was inferior to what was planned because of the COVID-19 pandemic, which required to temporarily shut down the screening unit and, as a consequence, to terminate the study prematurely. A strength of our study is that

it included only HPV-positive women in routine clinical practice. Second, the use of systematic biopsies and endocervical curettage for all HPV-positive women as standard reference regardless of the visual assessment interpretation avoided verification bias.

Replication with a larger sample size would be necessary to draw definitive conclusions. Nonetheless, most providers are convinced that the digital cervical image is useful for the diagnosis of CIN2+ lesions. Although the primary endpoint was inconclusive, this study represents the best available evidence to date that suggests that D-VIA/VILI may potentially improve cervical cancer screening.

Other avenues for improving cervical cancer screening exist, and new strategies to improve accuracy of visual assessment need to be investigated. The development of a smartphone application recognizing pathological lesions of the cervix, for example, is underway, and might contribute to improving the screening process [33].

## Conclusion

A digital approach performed in addition to naked-eye evaluation may be of clinically relevant benefit in CIN2+ detection. Indeed, the combination of VIA/VILI and D-VIA/VILI seems to provide an increase in sensitivity, with an acceptable decrease in specificity. Digital assessment is an innovative tool that may contribute to changing the current management of cervical cancer prevention in low-income countries. However, this will need to be confirmed by further studies with a larger sample size.

## Acknowledgments

The authors would like to thank the entire project team in Dschang and Geneva for their valuable contribution, and all the women who participated in the study in Cameroon. We would also like to thank all our funders for the economic support.

## Author Contributions

**Conceptualization:** Bruno Kenfack, Pierre Vassilakos, Patrick Petignat.

**Data curation:** Evelyn Tincho, Ania Wisniak, Jessica Sormani, Patrick Petignat.

**Formal analysis:** Eva Dufeil, Ania Wisniak, Patrick Petignat.

**Funding acquisition:** Bruno Kenfack, Jessica Sormani, Patrick Petignat.

**Investigation:** Patrick Petignat.

**Methodology:** Bruno Kenfack, Ania Wisniak, Jessica Sormani, Patrick Petignat.

**Project administration:** Jessica Sormani, Patrick Petignat.

**Resources:** Eva Dufeil, Ania Wisniak.

**Software:** Ania Wisniak, Jessica Sormani.

**Supervision:** Bruno Kenfack, Evelyn Tincho, Jovanny Fouogue, Jessica Sormani, Patrick Petignat.

**Validation:** Patrick Petignat.

**Visualization:** Patrick Petignat.

**Writing – original draft:** Eva Dufeil, Patrick Petignat.

**Writing – review & editing:** Eva Dufeil, Bruno Kenfack, Evelyn Tincho, Jovanny Fouogue, Ania Wisniak, Jessica Sormani, Pierre Vassilakos, Patrick Petignat.

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
