## [Decision Letter · Decision Letter 0]

7 Oct 2021

PONE-D-21-15964Addition of digital VIA to conventional naked-eye examination for triage of HPV-positive women: A study conducted in a low-resource settingPLOS ONE

Dear Dr. Dufeil,

Thank you for submitting your manuscript to PLOS ONE. After careful consideration, we feel that it has merit but does not fully meet PLOS ONE’s publication criteria as it currently stands. Therefore, we invite you to submit a revised version of the manuscript that addresses the points raised during the review process.

In particular, please address the reviewers' comments regarding additional details that should be presented in the manuscript. Also, please address in the methods, results, and conclusions how the overall results and interpretation of the trial were impacted by the trial being stopped short of its recruitment goal due to the COVID-19 pandemic.

We look forward to receiving your revised manuscript.

Kind regards,

Michael Scheurer, Ph.D.

Academic Editor

PLOS ONE

Journal Requirements:

Reviewers' comments:

Reviewer's Responses to Questions

**Comments to the Author**

1. Is the manuscript technically sound, and do the data support the conclusions?

Reviewer #1: Yes

Reviewer #2: Yes

2. Has the statistical analysis been performed appropriately and rigorously? 

Reviewer #1: Yes

Reviewer #2: I Don't Know

3. Have the authors made all data underlying the findings in their manuscript fully available?

Reviewer #1: Yes

Reviewer #2: No

4. Is the manuscript presented in an intelligible fashion and written in standard English?

Reviewer #1: Yes

Reviewer #2: No

5. Review Comments to the Author

Reviewer #1: Addition of digital VIA to conventional naked-eye examination for triage of HPV-positive women: A study conducted in a low-resource setting

Thank you for the opportunity to review this paper. It describes a study conducted in West Cameroon to evaluate the improvement in diagnostic performance when supplementing naked-eyed VIA/VILI with digital VIA/VILI using a low-cost smartphone camera. This low cost technology is exciting as it could potentially improve test-see-treat programs in low-resource settings by providing an inexpensive and readily accessible tool that may improve the sensitivity of VIA/VILI for triage of high-risk HPV positive patients. I have a couple of comments/suggestions for the authors:

1) Table 2 lists the VIA/VILI and D-VIA/D-VILI results of each of the 26 patients with CIN2+, stratified by CIN grade. The table is detailed in providing data at the participant level. However, it would be helpful to summarize the data for the reader. Would it be possible to present CIN grade- stratified 2x2 tables, indicating the number (%) within each cell? This would additionally provide the benefit of readily showing the concordance of naked VIA and D-VIA.

<cin2

VIA vs. D-VIA 2x2

CIN2

VIA vs. D-VIA 2x2

CIN3

VIA vs D-VIA 2x2

Cancer

VIA vs D-VIA 2x2

2) Is there any reason why data for patients with low grade CIN aren’t provided? Following the format suggested above, it would be helpful to have a 2x2 table for patients with <cin2.

3) Consort diagram and lines 210-213 on page 13. The authors indicate that 1 patient had a D-VIA with insufficient quality for assessment. However, the next sentence indicates that 188/188 patients with CIN2+ had D-VIA results, while 186 had VIA results.

4) It may be worth reporting on statistically significant differences in negative predictive value—one of the values of good screening/triage tests is providing reassurance to patients with negative triage tests that they can return in one year without fearing that they were mistakenly under-managed. While I didn’t do the math, it seems like a jump from 90.6% NPV to 94.7% is a significant improvement in the reassurance rate.

5) As the target sample size was understandably not attained due to COVID-19, it would be worth mentioning the power of the study given the actual sample size.

6) A few minor editorial comments:

a. Page 6, Procedure—“Midwives independently performed a naked-eye cervical evaluation (native, after acetic acid and Lugol’s iodine). Can you use i.e., after you use the word native to define what that means?

b. Page 4, Introduction, Line 74-76: “Advances in technology for ….. are adapted to the context.” It is unclear what is meant by “adapted to the context” (i.e., are contextual adaptations based on capacity and availability of resources/infrastructure?)</cin2.

</cin2

Reviewer #2: Summary: This study tested how well digital-VIA with a smartphone aided in diagnosis of cervical dysplasia when added to naked-eye VIA. These tests were compared to histological diagnosis of CIN2. The study had to be cut short because of COVID but provided some interesting data that suggests that D-VIA helped in diagnosis and same-day treatment of cervical dysplasia.

Major comments:

1. The idea of using a smartphone with digital images of the cervix to aid in diagnosis has the potential for a low cost and innovative improvement to traditional VIA.

2. The manuscript should be reviewed by a native-English speaker and editor as there are numerous sentences that could be written more clearly and spelling/grammatical mistakes could be corrected.

3. Who reviewed the VIA and D-VIA to determine if they needed treatment? Were these midwives? What if there was a difference of opinion between providers?

4. The methods section in regards to procedures needs to be expanded significantly: specifically, what were the questionnaires asking participants? When was biopsy and ECC being performed? Was this before or after VIA? Did everyone get VIA and VILI? How many images were taken? How were these stored or disposed of securely?

5. Table 1 only includes those who are HPV positive (188) but I think this really should be the total women recruited of 1081 and their characteristics.

6. Are the WHO recommendations for treating dysplasia only if CIN2+? Of those who were not treated as VIA and D-VIA missed dysplasia, how many were called back and successfully treated?

7. I would try and provide some more structure to the introduction: what is known about cervical cancer and its screening and treatment; what is not known; and then what you are hoping to accomplish. Some of the details of VIA and D-VIA could be included in the methods as your ‘intervention’.

8. Was regular VIA also completed prior to digital VIA? I imagine there could be some bias if digital VIA was done before naked-eye VIA.

Minor comments:

1. VIA is visual inspection with acetic acid and VILI is visual inspection with Lugol’s iodine. VILI has different test characteristics than VIA and so technically VIA should just include discussion of acetic acid and not also VILI.

6. PLOS authors have the option to publish the peer review history of their article (what does this mean?). If published, this will include your full peer review and any attached files.

Reviewer #1: No

Reviewer #2: No

---

## [Author Response · Author response to Decision Letter 0]

13 Mar 2022

Reviewer #1: Addition of digital VIA to conventional naked-eye examination for triage of HPV-positive women: A study conducted in a low-resource setting

Thank you for the opportunity to review this paper. It describes a study conducted in West Cameroon to evaluate the improvement in diagnostic performance when supplementing naked-eyed VIA/VILI with digital VIA/VILI using a low-cost smartphone camera. This low cost technology is exciting as it could potentially improve test-see-treat programs in low-resource settings by providing an inexpensive and readily accessible tool that may improve the sensitivity of VIA/VILI for triage of high-risk HPV positive patients. I have a couple of comments/suggestions for the authors:

1) Table 2 lists the VIA/VILI and D-VIA/D-VILI results of each of the 26 patients with CIN2+, stratified by CIN grade. The table is detailed in providing data at the participant level. However, it would be helpful to summarize the data for the reader. Would it be possible to present CIN grade- stratified 2x2 tables, indicating the number (%) within each cell? This would additionally provide the benefit of readily showing the concordance of naked VIA and D-VIA.

Thank you for this suggestion. We have summarized Table 2 according to your recommendations.

Table 2. VIA/VILI, D-VIA/VILI and combined assessment of histopathologic results, categorized by CIN grade

CIN grade

2) Is there any reason why data for patients with low grade CIN aren’t provided? Following the format suggested above, it would be helpful to have a 2x2 table for patients with 

We have added VIA/VILI and D-VIA/VILI assessment for negative and CIN1 cases in the table above.

3) Consort diagram and lines 210-213 on page 13. The authors indicate that 1 patient had a D-VIA with insufficient quality for assessment. However, the next sentence indicates that 188/188 patients with CIN2+ had D-VIA results, while 186 had VIA results.

We agree with this comment and there was indeed some confusion in these numbers. We have now excluded the non-interpretable case from the final analysis for D-VIA/VILI. In doing so, the sensitivity and negative predictive value of D-VIA/VILI have slightly changed. Changes have been made in tables 2 and 3, as well as in the manuscript.

4) It may be worth reporting on statistically significant differences in negative predictive value—one of the values of good screening/triage tests is providing reassurance to patients with negative triage tests that they can return in one year without fearing that they were mistakenly under-managed. While I didn’t do the math, it seems like a jump from 90.6% NPV to 94.7% is a significant improvement in the reassurance rate.

Thank you for this comment. P-values have been added for the difference in PPV and NPV between VIA/VILI and D-VIA/VILI and the combined test respectively, but showed no statistical significance at the p=0.05 level.

5) As the target sample size was understandably not attained due to COVID-19, it would be worth mentioning the power of the study given the actual sample size.

Thank you for this comment. We have added the current study power in the results section.

Lines 238: Considering our final sample of 26 CIN2+ participants, the current study was powered at 43.6% for the detection of a 20% difference between the sensitivities of VIA/VILI and the combined VIA/VILI and D-VIA/VILI method.

6) A few minor editorial comments:

a. Page 6, Procedure—“Midwives independently performed a naked-eye cervical evaluation (native, after acetic acid and Lugol’s iodine). Can you use i.e., after you use the word native to define what that means?

The following has been specified: 

Lines 131: Midwives independently performed a naked-eye cervical evaluation: native (i.e., product-free visualization of the cervix), after acetic acid and Lugol’s iodine.

b. Page 4, Introduction, Line 74-76: “Advances in technology for ….. are adapted to the context.” It is unclear what is meant by “adapted to the context” (i.e., are contextual adaptations based on capacity and availability of resources/infrastructure?)

We have completed this paragraph as follows:

Lines 74: Advances in technology for cervical detection in low-resource settings using HPV primary testing followed by immediate treatment with cryotherapy or thermal ablation if needed are adapted to the context (5)(6)(7). Indeed, a ‘screen-and-treat’ approach in a single visit requires fewer resources and reduces risk of loss to follow-up.

Reviewer #2: Summary: This study tested how well digital-VIA with a smartphone aided in diagnosis of cervical dysplasia when added to naked-eye VIA. These tests were compared to histological diagnosis of CIN2. The study had to be cut short because of COVID but provided some interesting data that suggests that D-VIA helped in diagnosis and same-day treatment of cervical dysplasia.

Major comments:

1. The idea of using a smartphone with digital images of the cervix to aid in diagnosis has the potential for a low cost and innovative improvement to traditional VIA.

We thank the reviewer for this comment.

2. The manuscript should be reviewed by a native-English speaker and editor as there are numerous sentences that could be written more clearly and spelling/grammatical mistakes could be corrected.

The manuscript has been reviewed and the grammatical mistakes have been corrected.

3. Who reviewed the VIA and D-VIA to determine if they needed treatment? Were these midwives? What if there was a difference of opinion between providers?

During the examination, one midwife independently assessed the need for treatment based on VIA/VILI and a second midwife independently based on D-VIA/VILI. A research assistant working on site ensured the quality of the data collected. In cases of disagreement, one positive assessment (either VIA/VILI or D-VIA/VILI) was sufficient to provide treatment. A gynecologist specialized in colposcopy reviewed all photos twice a month and gave her diagnosis based on D-VIA/VILI. Patients who were undertreated according to the specialist’s diagnosis were recalled for treatment.

The methods section has been completed accordingly:

Lines 131: Midwives independently performed a naked-eye cervical evaluation: native (i.e., product-free visualization of the cervix), after application of acetic acid and Lugol’s iodine. A second midwife independently captured images (native, after acetic acid and Lugol’s iodine) using a smartphone (Samsung Galaxy J5, Seoul, Korea). VIA and D-VIA were independently interpreted as non-pathological (no precancerous lesion) or pathological (suspicious of a precancerous or cancerous lesion). Decision to treat was based on the combined results (VIA and D-VIA); if one of the two tests (VIA or D-VIA) was positive, the decision was to treat. If the final decision was to treat, women were immediately treated by thermal ablation if eligible, or referred for further evaluation if non-eligible for thermal ablation. 

Smartphone photography allowed easy storage and retrieval of VIA images for quality control and educational purposes. Supervision of the screening activity was performed twice a month by a gynecologist specialized in colposcopy who reviewed all photos with the frontline health care providers. Patients who were undertreated according to the specialist’s diagnosis, as well as those with CIN2+ on histological analysis which were not identified through VIA/VILI or D-VIA/VILI, were called back for treatment. If the final decision was not to treat, women were invited for a follow-up in 1 year.

4. The methods section in regards to procedures needs to be expanded significantly: specifically, what were the questionnaires asking participants? When was biopsy and ECC being performed? Was this before or after VIA? Did everyone get VIA and VILI? How many images were taken? How were these stored or disposed of securely?

Thank you for this comment. The methods section has been completed with the following information.

Lines 123: After providing and signed informed consent, participants were administered questionnaires by a midwife on their socio-demographic characteristics, gynecological and obstetrical history, smoking status and HIV status.

Lines 128: If the HPV test was positive, a pelvic exam was performed with a PAP-test, followed by triage by VIA/VILI, and finally sampling by endocervical brushing and a biopsy at the site of pathological VIA/VILI or at 6 o’clock when VIA/VILI was negative.

Lines 133: A second midwife independently captured images during the pelvic exam (one native, one after application of acetic acid and one after application of Lugol’s iodine) using a smartphone (Samsung Galaxy J5, Seoul, Korea).

Lines 153: Smartphone images were anonymously stored on the server of the Geneva University Hospitals in a password-protected computer.

5. Table 1 only includes those who are HPV positive (188) but I think this really should be the total women recruited of 1081 and their characteristics.

Changes to Table 1 have been made according to the suggestion.

6. Are the WHO recommendations for treating dysplasia only if CIN2+? Of those who were not treated as VIA and D-VIA missed dysplasia, how many were called back and successfully treated?

Indeed, when histological results are available, the WHO recommends to treat only women with CIN2+ (WHO guidelines for treatment of cervical intraepithelial neoplasia 2-3 and adenocarcinoma in situ, 2014). In our study, all untreated participants in which CIN2+ was identified through histological analysis were called back and successfully treated. This has been added to the manuscript.

Methods - lines 148: Patients who were undertreated according to the specialist’s diagnosis, as well as those with CIN2+ on histological analysis which were not identified through VIA/VIL or D-VIA/VILI, were called back for treatment.

Results – lines 227: Both patients with CIN2+ identified by histological analysis who were missed by VIA and D-VIA were called back and subsequently treated.

7. I would try and provide some more structure to the introduction: what is known about cervical cancer and its screening and treatment; what is not known; and then what you are hoping to accomplish. Some of the details of VIA and D-VIA could be included in the methods as your ‘intervention’.

Thank you for this comment. We have restructured the introduction accordingly.

8. Was regular VIA also completed prior to digital VIA? I imagine there could be some bias if digital VIA was done before naked-eye VIA.

Regular VIA was performed independently by the first midwife. Pictures of the cervix were captured by the second midwife before application of acetic acid (native) and 60 seconds after application of acetic acid (VIA), which were not shown to the first midwife prior to her assessment. Lugol’s iodine was then applied for naked-eye assessment by the first midwife, before being photographed by the second midwife for independent digital assessment.

Minor comments:

1. VIA is visual inspection with acetic acid and VILI is visual inspection with Lugol’s iodine. VILI has different test characteristics than VIA and so technically VIA should just include discussion of acetic acid and not also VILI.

Thank you for this comment. VILI has been added throughout the manuscript when 

appropriate.

---

## [Decision Letter · Decision Letter 1]

21 Apr 2022

Addition of digital VIA to conventional naked-eye examination for triage of HPV-positive women: A study conducted in a low-resource setting

PONE-D-21-15964R1

Dear Dr. Dufeil,

We’re pleased to inform you that your manuscript has been judged scientifically suitable for publication and will be formally accepted for publication once it meets all outstanding technical requirements.

Kind regards,

Michael Scheurer, Ph.D.

Academic Editor

PLOS ONE

Additional Editor Comments (optional):

Reviewers' comments:

Reviewer's Responses to Questions

**Comments to the Author**

1. If the authors have adequately addressed your comments raised in a previous round of review and you feel that this manuscript is now acceptable for publication, you may indicate that here to bypass the “Comments to the Author” section, enter your conflict of interest statement in the “Confidential to Editor” section, and submit your "Accept" recommendation.

Reviewer #2: All comments have been addressed

2. Is the manuscript technically sound, and do the data support the conclusions?

Reviewer #2: Yes

3. Has the statistical analysis been performed appropriately and rigorously? 

Reviewer #2: Yes

4. Have the authors made all data underlying the findings in their manuscript fully available?

Reviewer #2: Yes

5. Is the manuscript presented in an intelligible fashion and written in standard English?

Reviewer #2: Yes

6. Review Comments to the Author

Reviewer #2: The authors have done a very nice job in addressing the reviewer comments. I just have a few minor comments but otherwise feel this manuscript should be accepted.

Minor comments:

1. In abstract, would not switch between presenting raw numbers or percentages first. I would pick one and use it continuously through the manuscript.

2. Second paragraph of introduction, first sentence could read for better clarity: “Cervical cancer screening allows for the detection and prevention of disease a precancerous stage…”

3. In discussion, the main limitation sentence should read: “The main limitation of the study…which required a temporary shut down of the screening unit, and as a consequence, to terminate the study prematurely.”

7. PLOS authors have the option to publish the peer review history of their article (what does this mean?). If published, this will include your full peer review and any attached files.

Reviewer #2: No

---

## [Editor Report · Acceptance letter]

4 May 2022

PONE-D-21-15964R1 

Addition of digital VIA/VILI to conventional naked-eye examination for triage of HPV-positive women: A study conducted in a low-resource setting 

Dear Dr. Dufeil:

I'm pleased to inform you that your manuscript has been deemed suitable for publication in PLOS ONE. Congratulations! Your manuscript is now with our production department. 

Kind regards, 

on behalf of

Dr. Michael Scheurer 

Academic Editor

PLOS ONE